# Rapid Evaporation of a Metal Electrode for a High-Efficiency Perovskite Solar Cell

**DOI:** 10.3390/polym16010094

**Published:** 2023-12-28

**Authors:** Runsheng Wu, Shigen Sun, Dongyang Liu, Junjie Lai, Yingjie Yu, Shijie Hu, Jun Liu, Shuigen Li, Yunming Li, Ling Li, Minhua Jiang, Chengyu Liu, Jun Deng, Chunhua Wang

**Affiliations:** 1School of New Energy Science and Engineering, Xinyu University, Xinyu 338004, China; 2Jiangxi Lithium Battery New Material Industrial Technology Institute, Xinyu 338004, China; 3Jiangxi Chunpeng Lithium Co., Ltd., Xinyu 338004, China; 4School of Energy and Environment, City University of Hong Kong, 83 Tat Chee Avenue, Kowloon, Hong Kong 999077, China

**Keywords:** perovskite solar cell, metal electrode, planar heterojunction, rapid evaporation

## Abstract

Organic-inorganic hybrid perovskite solar cells (PSCs) have attracted considerable attention due to the excellent optoelectronic properties of perovskite materials. The energy consumption and high cost issues of metal electrode evaporation should be addressed before large-scale manufacturing and application. We developed an effective metal electrode evaporation procedure for the fabrication of high-efficiency planar heterojunction (PHJ) PSCs, with an inverted device structure of glass/indium tin oxide (ITO)/poly[bis(4-phenyl)(2,4,6-trimethylphenyl)amine] (PTAA)/perovskite/[6,6]-phenyl-C61-butyric acid methyl ester (PCBM)/(E)-β-caryophyllene (BCP)/Ag. The effect of the evaporation rate for an evaporator with a small-volume metal cavity on the performance of PHJ-PSC devices was investigated systematically. Through controlling the processes of Ag electrode evaporation, the charge dynamics of the devices were studied by analyzing their charge recombination resistance and lifetime, as well as their defect state density. Our findings reveal that the evaporation rate of an evaporator with a small cavity is favorable for the performance of PHJ-PSCs. As a result, PHJ-PSCs fabricated using a very thin, non-doped PTAA film exhibit photoelectric conversion efficiency (PCE) of 19.21%, with an open-circuit voltage (Voc) of 1.132 V. This work showcases the great potential of rapidly evaporating metal electrodes to reduce fabrication costs, which can help to improve the competitiveness in the process of industrialization.

## 1. Introduction

In recent decades, PSCs have attracted considerable attention in the renewable energy field due to their high efficiency and low-cost processing potential [1,2,3,4,5,6,7,8]. Thus far, the PCE of PSCs has made great breakthroughs, from an initial value of 3.8%, reported by Miyasaka in 2009, to the latest certified value of over 26% [9,10]. This significant progress has arisen from the excellent physical and chemical advantages of metal halide perovskite materials, such as a high absorption coefficient, long charge diffusion, low exciton binding energy, long carrier recombination lifetime, and tunable direct bandgap [1,2,3,4,5,6,7,8]. The PCE also benefits from improvements and developments in the deposition methods of perovskite materials, selective extraction layers, and device structures [3,4,5]. To date, both inverted (ITO/PEDOT:PSS/perovskite/PCBM/Ag) and regular (FTO/compact TiO_2_/mesoscopic TiO_2_/perovskite/spiro-OMeTAD/Au) device architectures with p-i-n or n-i-p layouts have been applied in PSCs [3,4,5]. Moreover, many strategies, such as solvent engineering, interface engineering, etc., have been developed to minimize defects and increase the grain size, thereby improving the photovoltaic performance [11,12,13]. Additionally, to enhance the quality of perovskite films, many technologies, such as spin coating [14,15,16,17], spray coating [18,19,20], slot-die coating [21,22], vacuum deposition methods [23,24], and blade coating [25,26], have been adopted in practical applications for PSCs. Beyond these techniques, the electrodes of PSCs also play a crucial role in their performance. Many studies have focused on investigating the uniformity, physical mechanisms, and potential commercialization of PSCs by varying the electron-transporting layer, hole-transporting layer, perovskite photosensitizer, and their interfaces; however, reports on the electrode materials and their preparation methods for PSCs are still limited.

The cathodes or anodes of PSCs are usually composed of transparent conductive glass, where FTO is widely used for regular device architectures, while ITO is used for inverted ones due to its suitable work functions and heat resistance capability. Meanwhile, precious metal Au is employed in regular device architectures, while Ag, Al, and Cu are used for inverted architectures due to their suitable work functions [3,4,5,9,10]. Generally, the process of thermal evaporation is used in the production of metal electrodes in high-vacuum environments. Other technologies, such as inkjet printing, as well as screen printing and solution processes, have also been developed [27,28,29,30]. For the deposition approaches, Han et al. successfully prepared PSCs with a hole-transporting free layer by using carbon counter electrodes instead of noble metal electrodes, where a carbon black/graphite composite slurry was printed on the top of the ZrO_2_ layer using screen-printing methods, achieving PCE of 12.8% and long-term stability of 1000 h [31]. Planar PSCs with nanocarbon as counter electrodes and a hole extraction layer were developed for the first time by Yang et al. using an inkjet printing technique with a precisely controlled pattern and interface; the novelty of the design lies in the two-step formation of the perovskite layer and inkjet printing of the nanocarbon material [27], with PCE of 11.06%. Regarding the electrode, Henry and Cheng et al. demonstrated semitransparent planar heterojunction PSCs with comparatively high efficiency and a neutral color, where the opaque metal cathode was replaced by a either dielectric-metal-dielectric multilayer or a Ag nanowire [28,29,30]. Huang et al. prepared PHJ-PSCs with PCE values above 20% and good stability using a low-cost Cu electrode [32]. The powerful inkjet-printing, screen-printing, and solution process techniques, which allow for low-cost production, appear to be quite compatible with the electrodes of PSCs. However, these methods face significant challenges with respect to homogeneity and repeatability, as well as the large-scale preparation of PSCs using printing technology. It is also worth emphasizing that product consistency and repeatability play an important role in cost control for the commercialization of PSCs. Thermal evaporation technology, which is the mainstream technology for the preparation of metal electrodes of PSCs, precisely overcomes the drawbacks of the aforementioned deposition technology. This technique has the advantages of high repeatability and consistency; however, a high-vacuum environment is required when the metal electrode is evaporated, which consumes a considerable amount of evacuation time and electricity. Although studies have been conducted on the use of metal electrodes for PSCs, there are few reports on how to prepare metal electrodes more efficiently for PSCs using vacuum deposition. Therefore, it is of great scientific significance to prepare metal electrodes for PSCs with low costs and high efficiency, improving the competitiveness of their industrialization.

In this work, we focused on improving the metal electrode evaporation approaches to reduce the manufacturing costs and simultaneously enhance the PCE of the PHJ-PSCs. In particular, the influence of the evaporation rate on the PCE of PSCs was investigated systematically, with an analysis of the charge carrier dynamics of solar cell devices, such as the charge recombination resistance and lifetime, as well as the defect state density. Through a series of control experiments, we demonstrated that a metal electrode of PSCs prepared rapidly using an evaporation instrument with a small metal cavity was beneficial in boosting the performance of the PHJ-PSCs. Experimentally, we fabricated inverted PSCs with a device structure of ITO/PTAA/CH_3_NH_3_bI_3_/PCBM/BCP/Ag using a very thin, non-doped PTAA film, achieving maximum PCE of 19.21%, with a maximum Voc of 1.132 V. The method proposed in this work saves time in vacuuming and preparing metal electrodes, which can help to improve the production efficiency and enterprise competitiveness in the future for industrialization.

## 2. Experimental Section

### 2.1. Materials

Methylammonium iodide (CH_3_NH_3_I, 99%) and lead iodide (PbI_2_, 99.8%) were purchased from Xi’an Yuri Solar Co., Ltd. (Xi’an, China). The electron transport material fullerene derivative PCBM and chemical solvents (dimethyl sulfoxide (DMSO) and anhydrous N,N-dimethylformamide (DMF)) were purchased from Nano-Co., Ltd. (Westwood, MA, USA) and J&K Scientific Co., Ltd. (Beijing, China), respectively. The hole transport material PTAA was purchased from Sigma-Aldrich Co. LLC. (Tokyo, Japan). The precious metal electrode Ag (9.99%) and passivation material BCP were also purchased from Xi’an Yuri Solar Co., Ltd. (Xi’an, China). All chemicals and raw materials for PSCs were supplied by the commercial market and were used as received, without any further purification.

### 2.2. Preparation of PSCs

The PTAA and PCBM were dissolved in anhydrous chlorobenzene at concentrations of 2 and 20 mg/mL, respectively. To fully dissolve these two functional materials, they were heated and stirred for 12 h at 60 °C on a heating plate in a N_2_-filled glove box (H_2_O < 1 ppm, O_2_ < 1 ppm). BCP and CH_3_NH_3_I were dissolved in anhydrous chlorobenzene and absolute ethanol at concentrations of 0.5 and 30–40 mg/mL, respectively. PbI_2_ and CH_3_NH_3_I were dissolved in a mixture of anhydrous DMF and DMSO at a molar ratio of 1:4. After stirring and heating for 12 h at 70 °C, the color of the precursor perovskite solution changed from the initial turbidity to clear, indicating complete dissolution. All solutions were filtered using a 0.2–0.45 µm PTFE syringe filter before use to eliminate undissolved large particles.

The patterned ITO-coated glass substrates were sequentially cleaned in acetone, detergents, distilled water, and isopropyl alcohol for 10 min. After cleaning, these substrates were dried with clean nitrogen and treated with UV-ozone for 30 min. To prepare the hole transport layer, the PTAA solution was spin-coated onto the ITO/glass substrates at the speed of 5000 rpm for 30 s and dried on a hotplate at 100 °C for 10 min. The perovskite films were prepared by a two-step process. First, the perovskite precursor solution of PbI_2_ and CH_3_NH_3_I (with a molar ratio of 4:1) was dropped onto the PTAA-coated ITO/glass substrates and spin-coated at 4000 rpm for 40 s. During this process, the CH_3_NH_3_I solution was quickly dripped onto the center of the substrate after 10 s of spinning. Then, the wet perovskite film was heated on a hotplate at 100 °C for 10–30 min, resulting in a smooth and uniform CH_3_NH_3_PbI_3_ perovskite film. The electron transport layer and passivation layer were successively spin-coated onto perovskite-coated glass substrates using PCBM (dissolved in chlorobenzene at a concentration of 20 mg/mL) and BCP (dissolved in absolute ethanol at a concentration of 0.5 mg/mL). Finally, the silver electrode was evaporated at different speeds under a vacuum of ~4 × 10^−8^ pascals through a shadow mask. The illumination area of the PSC was designed to be 4.8–8 mm^2^.

### 2.3. Device Characterization 

The current density-voltage (J-V) curve of the PSCs was tested under a solar simulator (91160 s, Newport, AM 1.5 G) with different scan speeds from 0 V to 1.5 V under a light intensity of 100 mW cm^−2^. A standard silicon simulator was employed to calibrate the light intensity. Electrochemical workstations (CHI660D) were employed to assess the physical and chemical properties of the PSCs, such as the interface resistance, transmission resistance, and defect state. The crystallization of perovskite thin films was performed via X-ray diffraction (XRD, Rigaku D, Max 2500, Tokyo, Japan) with Cu Ka radiation with a scan rate of 8°/min from 10° to 50°, where the wavelength was 0.15406 nm and the scan mode was theta-2theta. A scanning electron microscope (SEM) was used to examine the morphology and crystallization of the perovskite thin films.

## 3. Results and Discussion

Considering that perovskite thin films are highly sensitive to oxygen, moisture, heat, ultraviolet light, and light soaking [33,34,35], the preparation of each functional layer of the PHJ-PSCs, including the electron/hole transport layers and passivation layer, was performed layer by layer through a solution process method in a nitrogen-filled glove box. Figure 1a illustrates the device structure of the PSCs, i.e., glass/ITO/PTAA/perovskite/PCBM/BCP/Ag, where the perovskite active layer was sandwiched between the hole and electron transport layers. The cathode of the PSC was completed by evaporating metal silver in a vacuum environment. The energy level of each functional layer is depicted in Figure 1b. It is evident that the energy level of PTAA matches that of the perovskite layer, creating effective hole contact that blocks electron extraction due to the shallow conduction band. Furthermore, due to its deep valence band, PCBM also serves as an excellent electron transport layer.

To obtain high-quality perovskite films, an improved two-step method has been developed. A precursor perovskite solution with excessive PbI_2_ was spin-coated on a heated PTAA-coated ITO substrate at a speed of 4500 rpm for 45 s. During this process, 60 μL of CH_3_NH_3_I isopropanol solution was dropped into the center of the PTAA-coated ITO substrate. Upon completion of spin coating, the wet perovskite-coated ITO substrate was immediately transferred to a hotplate at 100 °C for 30 min to facilitate the crystallization and growth of the perovskite film. The surface morphology and crystallization performance of the perovskite film are shown in Figure 2a,b. Numerous irregular voids were observed on the surfaces of the CH_3_NH_3_PbI_3_ perovskite and PbI_2_ films, regardless of the magnification. This could be attributed to the corrosion of the formed CH_3_NH_3_PbI_3_ perovskite and PbI_2_ compound film by the high-boiling-point DMF solution. Following the two-step deposition, by contrast, a high-quality perovskite film with a smooth and dense surface was formed, as exhibited in Figure 2c,d. This benefited from the loose and void PbI_2_ film, which enhanced the interdiffusion of PbI_2_ and CH_3_NH_3_I.

Using solution processing technology, a perovskite precursor solution with a molar ratio of 4:1 was used to prepare the CH_3_NH_3_PbI_3_ perovskite and PbI_2_ compound film. Figure 2e shows the XRD data of the CH_3_NH_3_PbI_3_ perovskite, represented by the black curve. The diffraction peaks at 14.37°, 20.27°, 23.74°, 24.82°, 28.72°, 32.13°, and 35.43° are assigned to the (110), (112), (211), (202), (220), (310), and (312) crystal planes, respectively, with lower diffraction intensity. The strongest intensity at 12.91° arises from PbI_2_, with the crystal plane of (001). Regarding the step-method-fabricated thin films, the XRD data of the CH_3_NH_3_PbI_3_ perovskite are shown in the red curve in Figure 2e. The diffraction peak positions and crystal planes are as follows: 14.37° and (110), 20.22° and (112), 23.74° and (211), 24.71° and (202), 28.72° and (220), 32.13° and (310), 35.21° and (312), 40.74° and (224), 43.29° and (314). The diffraction peak intensity is higher than that of the CH_3_NH_3_PbI_3_ perovskite and PbI_2_ compound film, indicating good crystallization, which is consistent with the SEM data.

Figure 3 depicts a schematic diagram of the evaporator and the evaporation method of the metal electrode, respectively. The evaporator consisted of a metal chamber, an evaporator boat, a molecular pump, a mechanical pump, and a sample stage. To prepare the metal electrodes of PSCs, the metal material was first placed on the evaporator boat, and the heat for evaporation was derived from the heating current. As the heat increased, the metal materials were transformed into a molten state. The molten metal then became a metal vapor and adhered to the surface of the BCP-coated ITO substrate. To ensure good contact between the metal electrode and the BCP passivation layer, air was removed from the metal chamber using mechanical and molecular pumps. The evaporation of the metal electrode can only occur when the pressure of the air in the metal chamber is below 8 × 10^−4^ Pascals. In this work, the metal chamber of the evaporation instrument was half the size of those used in other laboratories. Three preparation strategies, namely a conventional evaporation rate, high-speed evaporation rate, and ultra-high-speed evaporation rate, were adopted to achieve high-quality metal electrodes for PSCs. The detailed preparation process is shown in Figure 3b. As the thickness of the metal electron increased, the evaporate rate also increased. When the thickness of the metal electrode was below 50 nm, all three methods had rates below 0.6 nm/s. As the thickness of the metal electrode reached around 150 nm, the evaporation rate of the three evaporation methods steadily increased to 0.8 nm/s. However, when the thickness exceeded 180 nm, the conventional evaporation method had a rate of 1 nm, the high-speed evaporation method had a rate of 2 nm/s, and, for the ultra-high-speed evaporation method, it was 3 nm/s. During the process of electrode deposition, the thickness ranged from 200 to 1000 nm, with the evaporation rates of conventional evaporation, high-speed evaporation, and ultra-high-speed evaporation being 1, 2, and 3 nm/s, respectively. It is evident that the ultra-high-speed evaporation rate is three times faster than the conventional evaporation rate, which is beneficial for industrialization in the future.

Figure 4a displays the typical photovoltaic data of PHJ-PSC devices. The V_oc_ of the PSC devices varies under different deposition conditions: 0.99 V for conventional evaporation, 1.03 V for high-speed evaporation, and 1.04 V for ultra-high-speed evaporation. The short-circuit current densities (J_sc_) are 8.45, 21.89, and 23.48 mA/cm^2^, respectively. The filling factors (FF) of the three methods sequentially change from small (54.99%) to large (56.00%) and then to the maximum (72.90%). This leads to a similar change in the power conversion efficiency (PCE), which changes from 4.60% to 12.59% and then reaches the maximum value of 17.51%. Notably, the comprehensive factor of ultra-high-speed evaporation is higher than that of conventional evaporation. Figure 4b shows the Jsc versus voltage characteristics of the best PSCs, with a V_oc_ of 1.13 V, J_sc_ of 23.21 mA/cm^2^, and FF of 73.11%, yielding the maximum PCE of 19.21%. Additionally, Figure 4c displays the IPCE spectrum of the corresponding PHJ-PSC devices, while Figure 4d shows the absorbance spectrum of the CH_3_NH_3_PbI_3_ perovskite film. The IPCE data illustrate a favorable spectral response in the range of 300–850 nm, which aligns with the absorbance of the CH_3_NH_3_PbI_3_ perovskite film in Figure 4d. 

Figure 5 and Table 1 present the key statistical parameters of the PHJ-PSCs, i.e., the Voc, Jsc, FF, and PCE. Among the three electrode preparation conditions, the average Voc of the PSC devices is highest under ultra-high-speed evaporation, reaching 1.1 V, while it is less than 1 V for conventional evaporation and high-speed evaporation. Under ultra-high-speed evaporation, the average Jsc of PHJ-PSC devices exceeds 20 mA/cm^2^, while it is less than 10 mA/cm^2^ for the device with conventional evaporation. This difference could be attributed to the prolonged evaporation of the metal electrode, which causes heat to accumulate on the surface of the PHJ-PSC device. The longer the evaporation time, the more heat accumulates, potentially harming the PHJ-PSC device. The small metal chamber in the evaporator hinders heat diffusion. The FF and PCE of the PHJ-PSC device exhibit similar trends among the three electrode preparation conditions. The average PCE of the PHJ-PSC devices is highest under ultra-high-speed evaporation, reaching 16.9%, surpassing the conventional evaporation and high-speed evaporation. More detailed data are presented in Table 1.

Electrochemical impedance is a highly effective tool in analyzing carrier dynamics and is widely used in organic solar cells and lithium-ion batteries [36,37,38]. It also enables a deeper understanding of the physical and chemical properties of PHJ-PSC devices [39,40]. Figure 6a,b display the typical impedance spectra (IS) of PHJ-PSCs measured under different bias voltages (0.8 V or 0.9 V), without and with light irradiation conditions. The bias voltages of 0.8 and 0.9 V are very close to the V_oc_ of the PSCs, making them beneficial for reflecting on the working status of the solar cell. Therefore, only IS data with bias voltages of 0.8 and 0.9 V were collected. Notably, the presence of two distinct semicircles is observed for devices with light irradiation conditions, while only one semicircle is present for devices without light irradiation conditions. This suggests different transport processes in the PSCs at different frequencies. To extract valuable information, such as the carrier recombination resistance, carrier recombination lifetime, and defect state, the fitted curves of IS data are analyzed for PHJ-PSC devices with and without light irradiation conditions (Figure 7a,f). Importantly, regardless of the light irradiation condition, the recombination resistance for PHJ-PSCs with metal electrodes prepared by ultra-high-speed evaporation is higher than that of PHJ-PSCs with metal electrodes. This indicates that there is less electron-hole pair recombination in PSCs with ultra-high-speed evaporation, implying improved device performance. Additionally, the recombination lifetimes for PHJ-PSCs with metal electrodes prepared by ultra-high-speed evaporation are longer compared to those with metal electrodes prepared by conventional evaporation, regardless of the light irradiation condition. This finding aligns with the results of the recombination resistance. The defect state in PSCs significantly influences device performance. The trap density of states (tDOS) in PSCs can be calculated using the following formula [41,42].
N_T_(E_ω_) = −(V_bi_ω/qWk_B_T) × dC/dω(1)
E_ω_ = k_B_Tln(ω_0_/ω)(2)
where V_bi_, ω, q, W, k_B_, T, C, ω_0_ are the built-in potential, angular frequency, elementary charge, depletion width, Boltzmann’s constant, temperature, capacitance, and attempt-to-escape frequency, respectively. 

The defect state density of PHJ-PSCs with metal electrodes prepared by ultra-high-speed evaporation and conventional evaporation is approximately 1 × 10^13^ to 1 × 10^19^ m^−3^ and 1 × 10^14^ to 1 × 10^19^ m^−3^, respectively, as shown in Figure 7e. Similarly, under light irradiation (Figure 7f), the defect state density of PHJ-PSCs with metal electrodes, prepared using ultra-high-speed evaporation and conventional evaporation, is around 1 × 10^14^ to 1 × 10^18^ m^−3^ and 1 × 10^15^ to 1 × 10^18^ m^−3^, respectively. Therefore, it is evident that the defect state density of PHJ-PSCs with metal electrodes prepared by ultra-high-speed evaporation is lower than that of PHJ-PSCs with metal electrodes prepared by conventional evaporation, regardless of the presence of light irradiation. This finding reveals the superior photovoltaic performance observed in PSCs with ultra-high-speed evaporation. The lower defect state density can be attributed to reduced heat radiation from the heat sources during electrode evaporation, which is in good agreement with the results obtained for the recombination resistance and carrier recombination lifetime.

## 4. Conclusions

In summary, the effect of the metal electrode evaporation rate on the performance of PHJ-PSC devices using an evaporator with a small-volume cavity was investigated in this study. Specifically, taking inverted PHJ-PSCs with a device structure of glass/ITO/PTAA/perovskite/PCBM/BCP/Ag as an example, the relationship between the device performance and the electrode evaporation rate was studied. By uncovering how the evaporation rate affects the charge recombination resistance and lifetime and the defect state density of the PHJ-PSCs, we established a connection between the charge dynamics of PSCs and electrode evaporation. Our findings demonstrate that a faster evaporation speed in the evaporator with a small cavity leads to the improved performance of PHJ-PSC devices. Consequently, an inverted PHJ-PSC with the superior PCE of 19.21% and an excellent Voc of 1.132 V, fabricated using a very thin non-doped PTAA film, was achieved. The method of the ultra-high-speed evaporation of electrodes, as presented in this study, contributes to reducing the manufacturing costs of PSCs, which may offer benefits for their practical application.

## Figures and Tables

**Figure 1 polymers-16-00094-f001:**
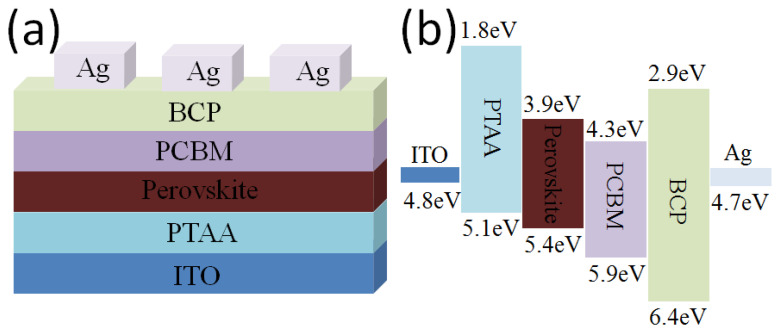
(**a**) The schematic of PHJ-PSC devices and (**b**) the energy level diagram.

**Figure 2 polymers-16-00094-f002:**
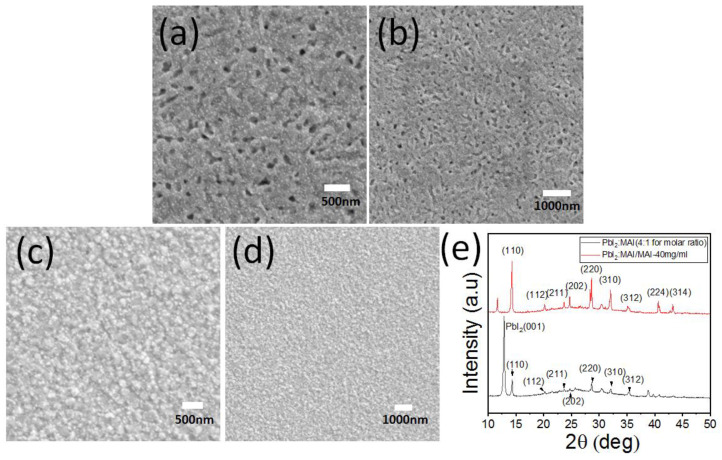
SEM at different magnifications of PbI_2_ (**a**,**b**) and CH_3_NH_3_PbI_3_ (**c**,**d**) perovskite thin films. (**e**) XRD images of CH_3_NH_3_PbI_3_ and compound perovskite thin films.

**Figure 3 polymers-16-00094-f003:**
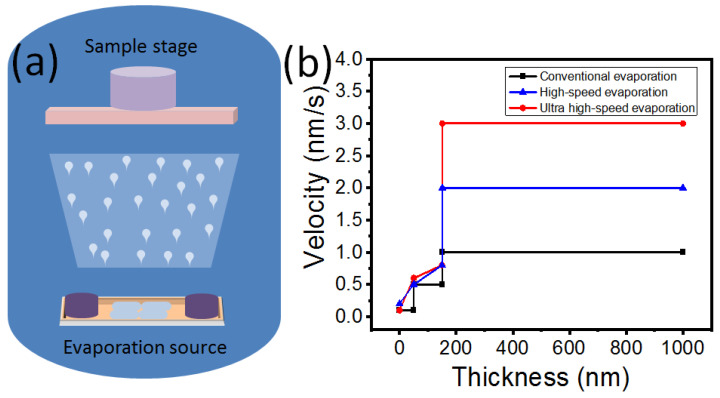
Schematic diagram (**a**) and preparation method (**b**) for metal electrode deposition.

**Figure 4 polymers-16-00094-f004:**
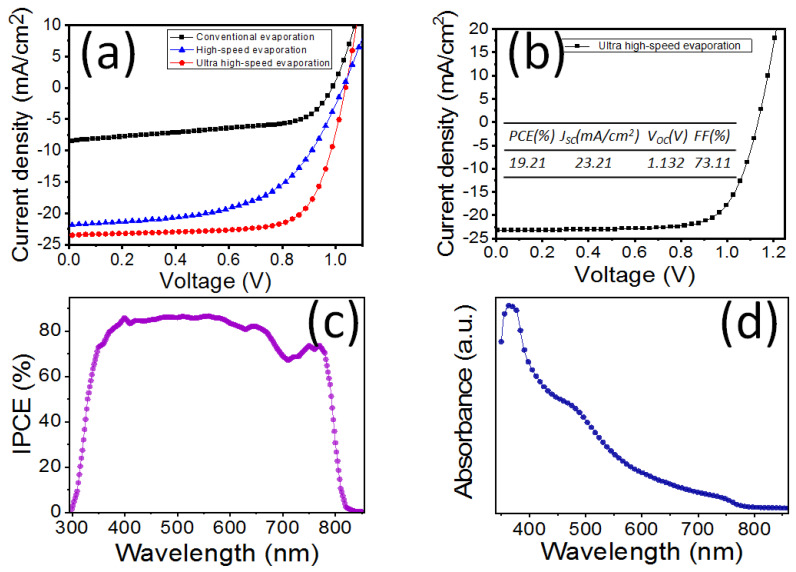
The J-V data of typical (**a**) and best (**b**) PHJ-PSC devices. (**c**) The IPCE spectrum corresponding to PHJ-PSC devices. (**d**) The absorbance spectrum of CH_3_NH_3_PbI_3_ perovskite film.

**Figure 5 polymers-16-00094-f005:**
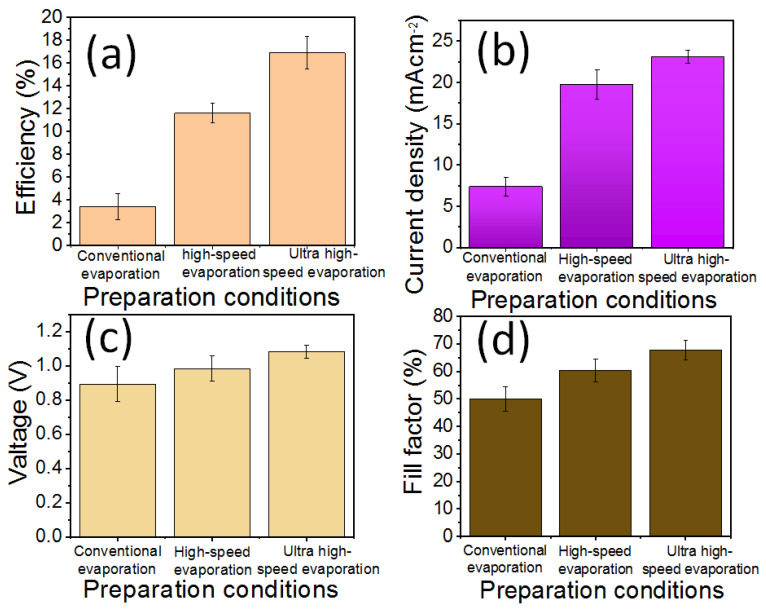
J-V data of PHJ-PSC devices under different evaporation methods.

**Figure 6 polymers-16-00094-f006:**
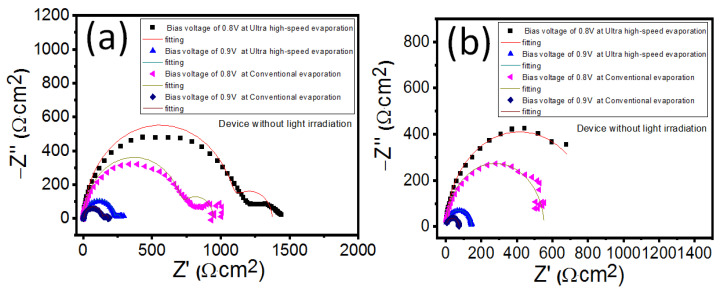
The impedance data of PHJ-PSC devices without (**a**) and with (**b**) light irradiation.

**Figure 7 polymers-16-00094-f007:**
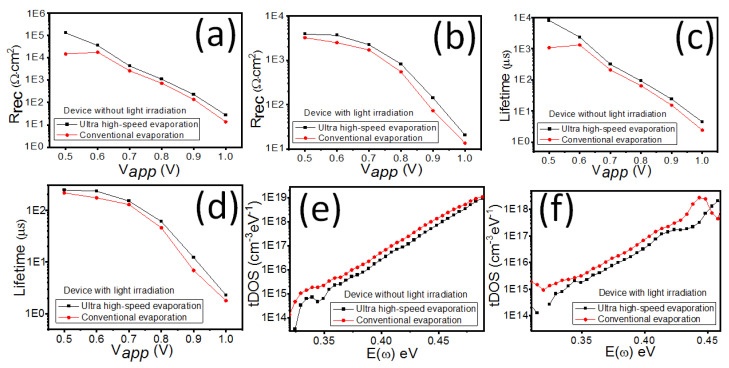
The recombination resistance (**a**,**b**) lifetime (**c**,**d**) and density of defect states (**e**,**f**) for PHJ-PSC devices without and with light irradiation.

**Table 1 polymers-16-00094-t001:** The J-V data of PHJ-PSC devices under different evaporation conditions (values in brackets represent the photovoltaic parameters for the best PSC device under corresponding evaporation conditions).

Condition	PCE (%)	Jsc (mA/cm^2^)	Voc (V)	FF (%)
Conventional evaporation	3.4 ± 1.2 (4.6)	7.4 ± 1.1 (8.45)	0.89 ± 0.10 (0.99)	50.0 ± 4.4 (55.0)
High-speed evaporation	11.6 ± 0.8 (12.6)	19.7 ± 1.8 (21.9)	0.98 ± 0.07 (1.0)	60.3 ± 4.1 (56.0)
Ultra-high-speed evaporation	16.9 ± 1.4 (19.2)	23.1 ± 0.8 (23.2)	1.1 ± 0.04 (1.1)	67.7 ± 3.7 (73.1)

## Data Availability

The authors confirm that the data supporting the findings of this study are available within the article.

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
