# Peer review of "Rapid Evaporation of a Metal Electrode for a High-Efficiency Perovskite Solar Cell"

_polymers, 2023, doi:10.3390/polym16010094_

Round 1

Reviewer 1 Report

Comments and Suggestions for Authors

The language requires substantial and strong improvements. a native English Speakers is needed. The discussion of the obtained results is very short and very week. The paper is not well done on both format and scientific contains. Based on the above remarks I recommend the reject of this paper.

Q1. The abstract and introduction part is not linked properly. Please revise the introduction part in line with the abstract.

Q2.

The novelty of the present work is not clear. There is lack of recent literature of the same issue and based on that, please elaborate the novelty of the present work.

Comments on the Quality of English Language

The language requires substantial and strong improvements. a native English Speakers is needed. 

Author Response

Q1. The abstract and introduction part is not linked properly. Please revise the introduction part in line with the abstract.

Response:.Thank you very much for your review. We have rewritten the introduction section of the manuscript and highlighted it in red.  

Q2. The novelty of the present work is not clear. There is lack of recent literature of the same issue and based on that, please elaborate the novelty of the present work.

Response: Thanks for reviewer’s careful review. We have rewritten the introduction, where novelty of the work is presented. We have cited the latest literature (2023) on some viewpoints. In the references section, we have highlighted it in red.  

Reviewer 2 Report

Comments and Suggestions for Authors

In this manuscript titled “Rapid evaporation of metal electrode for high-efficient perovskite solar cell”, authors developed a new method to reduce the thermal damage on the perovskite solar cell induced by metal electrode thermal evaporation. The experiment design was sound, and the results were well-interpreted. However, there are several items I would like to highlight before this manuscript can be accepted.

1.       Page5 Figure 3. Please specify in the figure caption that this is for metal electrode deposition, instead of “vacuum deposition for PSCs”.

2.       Page5 Figure 3 and all the following figures and tables, my understanding of the deposition speed was: conventional < high-speed < ultra high-speed. It would be more logical to follow the deposition speed sequence when making plots and tables.

3.       Page6 Figure 5, error bar is required on the bar chart.

4.       English writing requires major improvement.

In summary, I would recommend this manuscript be accepted after major revision.

Comments on the Quality of English Language

Needs major improvement 

Author Response

Q1. Page5 Figure 3. Please specify in the figure caption that this is for metal electrode deposition, instead of “vacuum deposition for PSCs”.

Response:Thank you very much for your review. We have corrected it.

Q2. Page5 Figure 3 and all the following figures and tables, my understanding of the deposition speed was: conventional < high-speed < ultra high-speed. It would be more logical to follow the deposition speed sequence when making plots and tables.

Response:Thank you very much for your review. We have corrected it. 

Q3. Page6 Figure 5, error bar is required on the bar chart.

Response:Thank you. The error bar has been added in Fig. 5. 

Q4. English writing requires major improvement.

Response: The major revisions in English have been implemented throughout the entire manuscript. 

Reviewer 3 Report

Comments and Suggestions for Authors

In the article entitled “Rapid evaporation of metal electrode for high-efficient perovskite solar cell” by Wu et al., the authors have presented a new way of doing metal electrode evaporation in planar heterojunction inverted perovskite solar cells by carefully controlling the evaporation speed and the small cavity. It addressed an interesting topic and showed the great enhancement in device characteristics with increasing electrode evaporation rate. Therefore, I think the manuscript is suitable for publication in Polymers after the following points have been addressed.

1.      Throughout the article, the authors talked about the great promise of rapidly evaporating metal electrodes to reduce fabrication cost. But doesn't using a faster evaporation rate consume more energy? 

2.     The authors should show EQE spectra of these devices along with the absorption data of the perovskite active layer.

3.     Unit of thickness is missing in Figure 3.

4.   What do the authors mean by ‘conventional evaporation’? If the conventional evaporation rate is similar to what other groups use in the research field, why does this cell show such low PCE?

5.     “Electrochemical impedance is a very effective tool in analyzing carrier dynamics and is widely used in organic solar cells and lithium-ion batteries.” References are missing here.

6.     Typos: ‘The band structure of each functional layer is shown in Fig. 1b. It can be seem that the energy band of PTAA matches one of perovskite layer, which is very suitable for hole transmission.’ It should be ‘seen’ not ‘seem.

Author Response

Q1. Throughout the article, the authors talked about the great promise of rapidly evaporating metal electrodes to reduce fabrication cost. But doesn't using a faster evaporation rate consume more energy? 

Response: Thank you very much for your constructive comments.

Fig.3 illustrates the schematic diagram of the evaporator and the evaporation method of metal electrode, respectively. The evaporator consists of a metal chamber, an evaporator boat, a molecular pump, a mechanical pump and a sample stage. When preparing the metal electrodes for PSCs, the metal material is first placed on the evaporator boat, in which the heat of  evaporation  boat comes from the heating current. With the continuous increase of heat, the temperature of metal materials keeps rising. When the temperature reaches the melting point of the metal material, the metal material will slowly melt and reach a molten state. As the heating continues, the molten metal will become metal vapor and adhere to the surface of the BCP-coated ITO substrate. To ensure good contact between the metal electrode and the BCP passivation layer, mechanical pumps and molecular pumps are used to remove air from the metal chamber. Usually, the metal electrode can only evaporate when the pressure of the air in the metal chamber is below 8*10-4 Pascals. In this work, the metal chamber of the evaporation instrument is half as large as that used in other laboratories. During the process of electrode deposition, the thickness ranges from 200 to 1000nm, and the evaporation rate of conventional evaporation, and ultra high-speed evaporation is 1 and 3nm/s, respectively. Clearly, ultra high-speed evaporation rate is three times the conventional evaporation rate. Due to the small metal cavity and and ultra high-speed evaporation, it saves time on evaporation and vacuuming. Hence, it reduce fabrication cost and is beneficial for industrialization in the future.  

Q2. The authors should show EQE spectra of these devices along with the absorption data of the perovskite active layer. 

Response: Thank you very much for your comments. We have supplemented this data.

Q3. Unit of thickness is missing in Figure 3.

Response: Thank you very much for your comments. We have revised it in the manuscript.

Q4. What do the authors mean by ‘conventional evaporation’? If the conventional evaporation rate is similar to what other groups use in the research field, why does this cell show such low PCE?

Response: Thank you. In our manuscript,the conventional evaporation rate is similar to what other groups use in the research field. But,the volume of metal chamber of the evaporation instrument is half as large as that used in other laboratories. The experimental results indicate that

indicate that the PCE of PSCs prepared using evaporation instruments with small metal cavities is very low. It can be attributed to the heat emitted by the evaporation boat on the surface of the solar cell.The evaporators with small metal chamber are not conducive to heat diffusion. Hence, the high-speed evaporation technology with a small metal cavity for metal electrode of PSCs was developted in our manuscript,which is beneficial for industrialization in the future. 

Q5. “Electrochemical impedance is a very effective tool in analyzing carrier dynamics and is widely used in organic solar cells and lithium-ion batteries.” References are missing here.

Response: Thank you very much for your comments. We have already cited references in the manuscript.

  1. Typos: ‘The band structure of each functional layer is shown in Fig. 1b. It can be seemthat the energy band of PTAA matches one of perovskite layer, which is very suitable for hole transmission.’ It should be ‘seen’ not ‘seem.

Response:Thank you. We have corrected it.

Round 2

Reviewer 2 Report

Comments and Suggestions for Authors

Authors have addressed my concerns, this manuscript can be accepted in the present form. 

Comments on the Quality of English Language

Can be improved 

Author Response

Last time, we have revised the manuscript according to the opinions of reviewer. He believes that we have completed his concerns and this manuscript can be accepted in the present form. Thanks for his constructive comments.